# Comparative Analysis Identifies Similarities between the Human and Murine Microglial Sensomes

**DOI:** 10.3390/ijms22031495

**Published:** 2021-02-02

**Authors:** Erik R. Abels, Lisa Nieland, Suzanne Hickman, Marike L. D. Broekman, Joseph El Khoury, Sybren L. N. Maas

**Affiliations:** 1Departments of Neurology and Radiology, Massachusetts General Hospital, NeuroDiscovery Center, Harvard Medical School, Boston, MA 02129, USA; eabels@mgh.harvard.edu (E.R.A.); lnieland@mgh.harvard.edu (L.N.); 2Department of Neurosurgery, Leiden University Medical Center, 2333 ZA Leiden, The Netherlands; m.broekman@haaglandenmc.nl; 3Center for Immunology & Inflammatory Diseases, Massachusetts General Hospital, Harvard Medical School, Boston, MA 02129, USA; shickman@mgh.harvard.edu; 4Department of Neurosurgery, Haaglanden Medical Center, 2597 AX The Hague, The Netherlands; 5Department of Pathology, University Medical Center Utrecht, 3584 CX Utrecht, The Netherlands

**Keywords:** microglia, RNAseq, sensome, aging, gene expression

## Abstract

One of the essential functions of microglia is to continuously sense changes in their environment and adapt to those changes. For this purpose, they use a set of genes termed the sensome. This sensome is comprised of the most abundantly expressed receptors on the surface of microglia. In this study, we updated previously identified mouse microglial sensome by incorporating an additional published RNAseq dataset into the data-analysis pipeline. We also identified members of the human microglial sensome using two independent human microglia RNAseq data sources. Using both the mouse and human microglia sensomes, we identified a key set of genes conserved between the mouse and human microglial sensomes as well as some differences between the species. We found a key set of 57 genes to be conserved in both mouse and human microglial sensomes. We define these genes as the “microglia core sensome”. We then analyzed expression of genes in this core sensome in five different datasets from two neurodegenerative disease models at various stages of the diseases and found that, overall, changes in the level of expression of microglial sensome genes are specific to the disease or condition studied. Our results highlight the relevance of data generated in mice for understanding the biology of human microglia, but also stress the importance of species-specific gene sets for the investigation of diseases involving microglia. Defining this microglial specific core sensome may help identify pathological changes in microglia in humans and mouse models of human disease.

## 1. Introduction

Microglia are the primary innate immune cells of the central nervous system (CNS). In addition to their protective function, these cells are involved in maintaining homeostasis in the brain parenchyma. Microglia are the main cells defending the brain against infections and are the primary resident inflammatory cells of the CNS [1]. Within a healthy brain, microglia have an important role in brain development and maintenance by clearing (cellular) debris. In pathological conditions, however, microglia can either limit disease progression or they can actually increase the burden of disease [2,3,4]. Recently, in addition to homeostatic microglia, a small subtype of microglia has been described based on their gene expression profile and role in disease progression, termed disease associated microglia (DAM) [4,5,6].

One of the main functions of microglia is to survey the environment to sense potential pathological conditions. In mice, the 100 most highly expressed genes responsible for sensing have previously been identified. This set of genes is termed the “sensome” and was first described by Hickman et al. [7,8]. These 100 genes include purinergic receptors, cytokine receptors, chemokines, Fc receptors, pattern recognition receptors, extracellular matrix (ECM) receptors, endogenous ligands receptors, sensors, transporters, and proteins involved in cell–cell interactions. The expression of the murine sensome genes decreases during aging and has recently also been reported to be downregulated in the presence of a glioma [3]. In addition, downregulation of the sensome genes is observed directly after traumatic brain injury (TBI), with expression being restored to baseline over time [9]. This indicates that aged microglia and microglia in the presence of a neoplasm, or after TBI, have a decreased capacity to sense their environment, possibly leading to reduced host defense response. Here, we have analyzed and discussed a number of microglia bulk and single cell RNAseq datasets from mouse and human disease models and found a number of sensome genes downregulated over time or over the course of disease [4,10,11,12,13]. The decrease in sensome expression could result in accelerated neurodegeneration of the brain or in the case of an existing neoplasm in enhanced tumor progression [3,7,9].

So far, the microglial sensome has been defined by only one study in mice [7]. The aims of this study were to determine if analysis of different datasets will result in similar sensomes and to identify similarities and differences between the mouse and human sensomes defined as a microglia specific set of genes encoding proteins that perceive extracellular signals. Examples of these extracellular signals include infectious pathogens, amyloid peptides, or tumor cells or their derivatives. In these pathological settings, the ability of microglia to sense changes in their environment may alter disease progression [14]. Because of inherent difficulties in comparing datasets generated from different sources using different methodologies and to facilitate our comparisons of the human and mouse transcriptomes, we re-analyzed the mouse and human transcriptomes generated by Gosselin et al. using the same approach first used by Hickman et al. to generate the initial sensome data by including automated screening and manual curation of every candidate gene. We compared the Gosselin et al. dataset for similarities with the initial set of sensome genes and compared their mouse and human datasets. We then performed the same analysis on an additional human dataset (Galatro et al.). There was significant overlap between the two mouse and human microglial sensing gene sets [15,16]. After identifying the sensome genes in both human and mouse microglia, we were able to identify a set of 57 genes that were present in at least three out of four extracted sensomes, which we termed the “microglia core sensome”. We then tested the usefulness of this core sensome by examining the pattern of changes in these genes from five different single cell and bulk RNASeq datasets from various conditions and disease models including Alzheimer’s disease (AD), aging, and amyotrophic lateral sclerosis (ALS). Defining the microglial core sensome genes may help identify common as well as differential pathological changes in microglia in humans and mouse models of human disease and would allow for a more focused tracking of changes in these cells during disease progression.

## 2. Results

### 2.1. Identification of Microglial Mouse Sensome Shows Major Overlap between Two Independent Datasets

The microglial sensome as described by Hickman et al. is a set of genes highly expressed in microglia that identifies the top 100 genes used for sensing of their surrounding [7]. To investigate whether we could validate and update this set of genes, we used the same approach as Hickman et al. and applied it to another published microglia transcriptome dataset [16]. Using this dataset, we set out to identify a number of genes that are potential candidate sensome genes. Microglial RNAseq data from Gosselin et al. provide transcriptome information from microglia as well other cells from the mouse cerebral cortex together, similar to Hickman et al. [7,16]. In Gosselin et al., mouse microglia were isolated after gentle mechanical dissociation following fluorescence activated cell sorting (FACS) identifying microglia as live/DAPI–, CD11B^+^, and CD45^low^ single cells, whereas Hickman et al. used a combination of mechanical and enzymatic dissociation following microglia isolation using FACS and the microglia were defined by FACS, collecting CD45^mid^ and CD11B^high^ microglial cells [7]. To select the sensome genes, we applied sequential criteria to filter the dataset. First, we performed a technical quality control of the samples and reads. Second, microglia specific genes were determined at a cutoff of log_2_fold ratio >2 in microglia versus cortex (all cells in cortical brain tissue). This cutoff was chosen to mimic the approach of the determination of the original sensome by Hickman et al. These criteria retrieved genes highly enriched in microglia. Only genes with an FDR-adjusted *p*-value <0.05 were included. Third, the microglia specific genes were annotated and mapped by filtering the genes based on associated GO terms. These included “plasma membrane”, “integral component of membrane”, “integral components of plasma membrane”, and “transmembrane signaling receptor activity”. We used these criteria to select genes that are translated into proteins only present on the plasma membrane, because proteins involved in sensing can only be found on the plasma membrane, as originally described [7]. Finally, the top 25% highest expressed genes in microglia were selected to create a set of candidate sensome genes (Figure 1A). Using our criteria, true sensome genes are defined as microglia specific genes, transcribing plasma membrane localized proteins involved in the binding of extracellular ligands. By comparing the top 100 (Figure 1B) and all 576 candidate sensome genes extracted from the publication of Gosselin et al., it is clear that many of the original sensome genes defined by Hickman et al. are among the identified candidate genes (Appendix A). However, our approach also yielded genes that may be associated with the plasma membrane as defined by the GO terms used, but that exert extracellular functions rather than the activation of intracellular pathways (e.g., Tnf). Therefore, we manually screened all candidate genes until a selection of 100 sensome genes were identified by three independent manual screens, and termed the “Gosselin mouse sensome”. By plotting the overlap between the genes found in our newly defined data selection, including the top 100 sensing genes from Gosselin et al. and the original sensome from Hickman et al. (Appendix A), we found a 73% overlap between the two independent murine sensome extractions (Figure 1C). These overlapping genes were termed the “mouse core sensome” with another set of 127 genes that include the genes uniquely present in either one of the extractions termed the “mouse extended sensome”. To determine the function of the mouse sensome overlapping genes, we created a circ plot, where the genes were ordered by differential expression based on microglia versus cortex from the publication of Gosselin et al. [16]. Every gene was categorized into eight different groups. These include purinergic receptors, cytokine receptors, chemokine and related receptors, Fc receptors, pattern recognition and related receptors, ECM receptors, endogenous ligands receptors, sensors and transporters, proteins involved in cell–cell interactions, and potential sensors with no known ligands. We found that the majority of the genes are involved in pattern recognition, which is crucial to the innate immune response recognizing infectious organisms and other pathogenic ligands (Figure 1D). The remaining genes were more evenly distributed among the other seven groups. As the publication of Gosselin et al. features RNA expression data on both human and mouse microglia, this publication includes a normalized expression profile of each gene comparing the expression in mice to humans. Analysis of the expression of the mouse sensome overlapping genes in this dataset found a distribution of genes that are higher expressed in mice compared with humans (Figure 1E).

### 2.2. Identification of the Human Microglial Sensome

To define the human microglial sensome, we used two independent published datasets (Gosselin et al. and Galatro et al.), where RNA expression levels from both human microglia and cortex were included [15,16]. Gosselin et al. isolated microglia from brain samples removed from 19 patients with epilepsy, serving as control, brain tumors, and acute ischemia. Here, mechanical dissociation was followed by microglia isolation using FACS and microglia were defined by live/DAPI^-^, CD11B^+^, CD45^low^, CD64^+^, and CX3CR1^high^ cells. Within these samples, no strong patterns related to diagnosis or age were detected. Galatro et al. collected microglial samples and corresponding parietal cortex tissue from autopsy specimen 6–24 h post-mortem from donors without any diagnosed brain diseases. Here, tissue was also mechanically dissociated following microglia isolation using FACS, where microglia were defined by live/DAPI^−^, CD11B^high^, and CD45^int^ single cells. In total, 39 microglial and 16 parietal cortex samples were collected. With the published human microglia data, we used an approach similar to the one used in mice; (1) the results were filtered for technical quality control, (2) followed by differential expression, (3) mapping to GO terms, and (4) filtering of the top 25% genes in microglia. This screen resulted in 506 candidate sensome genes from the Gosselin dataset and 525 candidate sensome genes from the Galatro dataset (Figure 2A). Similar to our approach in the murine datasets, all candidate genes were manually checked and filtered until 100 sensome genes were selected. From these sensome sets, we detected an overlap of 75% between the two sensomes identified from the independent datasets (Figure 2B and Appendix A). The 75 overlapping genes were termed the “human core sensome” and the combined set of 125 overlapping and non-overlapping genes were termed the “human extended sensome”. To get an insight into the function of these genes, we assigned the genes according to the various groups used to distinguish the gene functions of the mouse sensome (Figure 2C). Interestingly, in the human sensome, a relatively higher number of genes were part of the ECM receptors and endogenous ligands receptors, sensors, and transporters genes groups compared to what we found in mice. Overall, however, we found that the genes were equally distributed over the different groups between mouse and human (Appendix A). The distribution of the genes in the human sensome was evenly distributed between human- and mouse-enriched genes (Figure 2D).

### 2.3. Mouse and Human Microglia Express a Core Set of Sensing Genes

We then determined the overlap between the mouse and human sensome genes by comparing the mouse sensome to the murine orthologs of the human sensome. To identify a core set of microglial sensing genes, present in both humans and mice, we selected the genes that were present in at least three out of four extracted sensomes. A total of 57 genes matched these criteria (Figure 3A). The identification of the 57 overlapping genes between the mouse and human sensomes shows that microglia from human and mice use a common set of 57 core genes to sense their environment (Figure 3A). These microglia core sensome genes can be divided into different groups and include genes of all eight functional domains, including purinergic receptors, cytokine receptors, chemokine and related receptors, Fc receptors, pattern recognition receptors, ECM protein receptors, proteins involved in cell–cell interactions, and sensors or transporters (Figure 3B).

Next, we set out to analyze what specific ligands were recognized by the identified sensome genes. This was achieved by manually curating which ligands were recognized per receptor using the database Uniprot (https://www.uniprot.org/) (Appendix A). First, we analyzed the overlap between the human and mouse ligands recognized by the human and mouse sensome genes. We performed this analysis to probe if the different genes expressed by human and mice recognize similar ligands. Here, we did find an overlap between the human and mouse ligands recognized (Appendix A). Second, to further generalize the ligands recognized by the receptors, we categorized all ligands in specific ligand groups, including glycoproteins, cytokines, immunoglobulin, amino acids, carbohydrates, electrolytes, lipopeptides, chemokines, neuraminic acids, nucleic acids, receptors, lipids, fatty acids, leukotrienes, hormones, steroids, and phospholipids (Appendix A). Again, we found that the mouse and human microglia can sense, in general, the same groups of ligands (Appendix A).

In conclusion, by comparing different datasets from both human and mouse, we identify a core set of sensome genes (Figure 3). These genes include microglial marker genes, such as TMEM119 [17,18,19] and several genes involved in the pathogenesis of CNS disorders such as AD (CD33, CX3CR1, and TREM2) and hereditary diffuse leukoencephalopathy (CSF1R) [4,20,21,22,23,24]. Overall, we define a set of genes important in sensing by microglial cells in both human and mouse, which we term the “microglia core sensome” (Figure 4 and Appendix A).

### 2.4. Changes of Sensome in Different Models

As we established a core sensome in this study, we extended our analysis to investigate if the sensome is changed during the course of disease or during aging. We analyzed different existing datasets that included data on AD, aging, and ALS (Appendix A). We found various changes in the microglia core sensome identified in AD (Figure 5), aging (Appendix A), and ALS (Appendix A) in murine and human microglia. These datasets included different mouse models used in AD datasets that examined the disease using different mouse models (familial AD gene mutations (FAD)), calcium/calmodulin-dependent protein kinase II α (CaMKII) promoter (CK-p25), amyloid precursor protein (APP), and AD-transgenic (Tg-AD) mouse models) [4,12,13]. During AD, Trem2, Fcgr4, Itgam, Mpeg1, Slc11a1, and Slc16a3 were upregulated, while P2ry12, P2ry13, Parvg, Selplg, Slc2a5, Slco2b1, Tmem119, and Tnfrsf1b were found to be upregulated (Figure 5). We also analyzed gene expression data derived from mouse models used in aging studies (including the ERCC1 mouse model, where DNA repair is impaired) [10,12,15] and models simulating ALS (SOD1G93A mutant versus SOD1WT and SOD1 versus control) [11,12]. Interestingly, we found that Clec7a and Tlr2 were significantly upregulated during aging, ALS disease progression, and AD. We also found disease-specific gene changes. In ALS, P2ry12 and P2ry13 were found to be downregulated (Appendix A). We organized all the data extracted from the different studies in the Appendix A. Overall, these data show that the capacity of microglia to sense changes in their surroundings changes in the course of neurodegenerative diseases and during aging.

## 3. Discussion

In this study, we sought to validate and extend the published mouse microglia sensome [7], define the human microglia sensome, and determine which sensing genes are highly expressed in both species. Using published datasets, we found a 73% overlap between the previously identified mouse microglia sensome and an independently extracted mouse sensome set. This set of 73 genes was termed the “mouse core sensome”. We applied the same methodology to published human microglial RNAseq data to identify genes that make up the human microglial sensome. The mouse samples from the different studies were freshly isolated using different dissociation techniques and antibody panels to identify microglia. The human samples were isolated from post-mortem CNS tissues from neurologically healthy donors (Galatro et al.) and from neurosurgical brain tissue of neuropathological patients (Gosselin et al.), where microglia were defined using different markers between the studies [15,16]. For both the mouse and human source datasets, different RNA extraction and sequencing techniques were used, yet these datasets yielded a 73% and 75% overlap in sensome genes within each species, respectively. Importantly, we found overlap between human and mouse sensome sets. Based on our analysis of the four sensome sets, we identify a set of microglial sensing genes that are highly expressed in both species and termed this the “microglia core sensome”. Utilization of these microglia core sensome genes in subsequent research may help identify pathological changes in microglia in both humans and mice.

To test the usefulness of our core sensome in various CNS disorders, we extended our analysis to include five additional datasets and compare the changes observed in sensome in these datasets. Interestingly, we found that similar changes occur in different datasets of the same disease or condition (such as AD), but that different changes occur when comparing different diseases such as AD and ALS. Interestingly, Tlr2 was found to be upregulated in all disease models. This was previously reported in different models studying Tlr2 expression of microglia during inflammation and neurodegenerative diseases [25]. In this analysis into the different diseases, Parkinson’s disease was not included. This disease model has not been studied to such an extent compared with the other neurodegenerative disease analyzed here. Future studies into Parkinson’s disease and specifically into microglia can shed light onto the role of these cells and how they are affected during disease progression. Prior to our analyses, changes in the original sensome genes defined by Hickman et al. had already been described for different models. For example, in an experimental model of TBI, it has been shown that the sensing capabilities of microglia change over time post TBI. From the 46 sensome genes analyzed, the majority were downregulated 2 days post injury. The expression normalized over time, returning to baseline expression 14 days post TBI [9]. In another study in experimental autoimmune encephalomyelitis (EAE), a model for multiple sclerosis (MS), it was found that microglial sensing was dysregulated. This receptor binds tumor necrosis factor alpha (TNFα) and activates anti-inflammatory and neuroprotective pathways [26]. Using a mouse model with a floxed Tnfrsf1b gene in combination with tamoxifen-inducible CRE expression driven by the microglial CX3CR1 promoter (Cx3cr1CreER:Tnfrsf1bfl/fl), 46% of the sensome genes were downregulated after knock-out of the Tnfrsf1b gene. This was accompanied by a reduced capacity for phagocytosis [27]. This shows that, in the case of neurological damage, the microglial sensome is dysregulated. However, in a model of progressive myoclonus epilepsy of Unverricht–Lundborg type, characterized by the loss-of-function mutation of cysteine protease inhibitor cystatin B (CSTB), microglia were found to be in an activated state as measured by increased expression of interferon-regulated genes. When examining the gene expression of 81 sensome genes, no difference was found in gene expression compared with control. Most changes were found in genes regulated by interferon, suggesting that, in this disease model, the microglial sensing function is not affected [28]. As opposed to other studies describing differences in sensome expression, this study [28] used in vitro cultured microglia instead of fresh ex vivo isolated microglia [7,9,27,28]. Other models have been used to study microglia function in peripheral viral infection in piglets, where it was found that expression of sensome genes was induced [29]. This supports the conclusion that, upon viral infection, microglia increase the expression of sensing genes in order to detect and combat the infection. In addition, it demonstrates that expression of various sensome transcripts changes in response to infection in additional species other than mice or human.

When we compared the mouse and human sensomes and found overlap between the gene sets, some technical differences between the studies may have influenced our findings and reduced the level of overlap. Differences between the neurological status of the tissues used to comprise the human and murine sensome could have affected the selection of sensome genes, as murine tissue was collected from healthy mice and the human tissue was collected from neuropathological tissue or autopsy material. Moreover, different studies use different isolation methods (antibody panels and tissue dissociation techniques, which can potentially change microglia gene expression or select for subpopulations of microglia). While the studies used these different techniques, we were able to find a significant overlap.

Our core sensome set was identified from datasets that utilized bulk RNAseq of pooled microglia compared with bulk RNA Seq of whole brain to assess for cell enrichment, similar to the way in which the original microglia sensome was defined [7]. With single cell microglia datasets now available, we checked if similar results could be obtained from single cell microglial RNA. To do this, we identified the top 100 expressed sensing genes from mouse homeostatic microglia as published by Keren-Shaul et al. [4]. Even though this pipeline was unable to enrich for microglial expression by comparing to whole brain RNA, 49 out of the 100 genes were shared with the sensomes extracted by Hickman et al. and by us from the Gosselin et al. dataset (Appendix A). Overall, 61% of core sensome genes were part of the single cell derived microglial sensome (Appendix A) [4].

Even though a number of murine models are being used to study human neurological diseases, various differences in gene expression between mouse and human microglia have been found [30]. A discrepancy in aging was previously described by Galatro et al., however, still using the murine dataset [15]. In this study, it was further described that a core microglia signature shows extensive overlap between mouse and human, but when studying this signature during aging, a species-specific age-related divergence appears [15]. Another example of the divergence between human and mouse microglia is present in the amyloid response during AD. Whereas in mice, during amyloid formation, homeostatic gene expression is downregulated and a disease-associated microglia (DAM) gene signature is increased [4], the opposite is found in homeostatic gene expression in human microglia in response to amyloid [31]. An increased insight into the difference between human and mouse microglia is found using single cell transcriptomics; this has advantages in understanding the different subsets of microglia in both mouse and humans. Surprisingly, different clusters of microglia were identified in humans, characterized by the expression of CCL2, CCL4, EGR2, and EGR3. It has been hypothesized that these clusters resemble a more activated state of microglia. The discrepancy in the microglia clusters is possibly due to differences in environmental factors, such as the hyper-hygienic animal facilities wherein mice are kept [14,32,33]. In addition, the mice strains are similar in their (epi)genetic composition, while the human donors are much more diverse both in their exposure to their environment as well as due to their (epi)genetics. This discrepancy between human and murine microglia is also shown in a cross-species single-cell analysis, which revealed numerous differences between human and murine microglia. While microglial core genes (e.g., TMEM119 and P2RY12) were found to be conserved, differences between human and mouse microglial gene expression were apparent in genes responsible for phagocytosis, complement, and susceptibility to neurodegenerative diseases [34,35]. Overall, these data indicate that differences between mouse model and humans need to be taken into account when translating the results acquired from murine models to humans.

Murine and human microglia can also respond in a similar way when confronted with pathological conditions. Recently, we showed that both the mouse and human microglial sensing potential is reduced in the context of glioma [3]. This suggests that microglia in the context of a tumor have a reduced capability to sense danger and can thus not fully execute their host-versus-tumor response. We hypothesize that the tumor, owing to the altered expression of specific microglial genes, can thus further thrive. This finding underscores the value of this set of genes in physiologic and pathological conditions.

In conclusion, we confirmed the robustness of the mouse sensome and identified core and extended mouse and human sensomes that can now be applied to further study microglia in physiologic and pathologic settings. While we established a list of genes that are involved in sensing in both human and mice, this needs to be validated using different techniques (such as immunohistochemistry, CyTOF, or CITEseq) targeting a number of these proteins (e.g., CD33, CX3CR1, P2RY12, and TMEM119) to see if the microglia also express the proteins encoded by these genes [8]. In addition, we identify a set of 57 genes that are important for specific microglial sensing in both human and mouse, which we termed the “microglia core sensome”. Understanding the dynamic and rapid changes microglia undergo in these different settings can help find therapeutic avenues in diseases where microglia play a key role, such as in cancer, TBI, and neurodegenerative diseases. Most importantly, with this shared set of sensing genes identified, future studies can use this information to study these genes in mice for alterations that are transferable to human microglia. This may help identify new druggable targets to reverse the changes observed to microglia sensome genes in different neuropathological processes.

## 4. Materials and Methods

### 4.1. Data Sources

For the extraction of the (overlapping) murine and human sensomes, published datasets were used from the publications of Hickman et al., Gosselin et al., and Galatro et al. [7,15,16]. For the application of the microglia core sensome on (murine and human) microglial aging and disease data, the published (bulk and single cell) RNAseq data from multiple studies were used [4,10,11,12,13,15]. As a validation step using murine single-cell microglia, the published data from the Keren-Shaul et al. publication was used [4].

### 4.2. Data Analysis and Pre-Processing

All data analysis was performed in R 3.3.3. Samples with less than 6000 genes with at least five mapped reads were excluded from analysis (*n* = 0). For the generation of heatmaps, read counts were normalized using the regularized logarithm transformation method from the DESeq2 R package [36]. The ligands for all receptors were determined by datamining the Uniprot database (https://www.uniprot.org/). Hereafter, the ligands were categorized into specific groups (glycoproteins, cytokines, immunoglobulin, amino acids, carbohydrates, electrolytes, lipopeptides, chemokines, neuraminic acids, nucleic acids, receptors, lipids, fatty acids, leukotrienes, hormones and steroids, and phospholipids) (Appendix A).

### 4.3. Extraction of Sensome Genes

All published datasets were downloaded from the published repositories or the supplementary data files. The raw datasets were loaded and analyzed using the DESeq2 R package [36]. To select for genes specifically expressed in microglia, candidate genes were identified as log_2_fold enrichment of >2 and a false-discovery rate (FDR) adjusted *p*-value of <0.05 comparing microglia versus whole brain gene expression. To select for genes coding for proteins that are located at the plasma membrane and function in a receptor-like manner, we used a pre-screen by selecting gene ontology (GO) mapping for the microglia specific selected genes. This was performed by using the biomaRt R package using the hsapiens_gene_ensembl or mmusculus_gene_ensembl datasets [37]. All genes that included the GO terms “plasma membrane”, “integral component of membrane”, “integral component of plasma membrane”, or “transmembrane signaling receptor activity” were included. All genes that did not map to any GO terms were manually screened using the Uniprot database (https://www.uniprot.org/). Next, similar to the identification of the original mouse microglial sensome by Hickman et al., the top 25% expressed microglial genes were deemed candidate sensome genes. All candidate sensome genes were independently checked by three investigators (E.R.A., L.N., and S.L.N.M.) for applicability (i.e., the capacity to detect extracellular signals and propagate this intra-cellularly) until 100 genes were selected by all three investigators. This manual curating of candidate genes was done using the Uniprot database supplemented with PubMed. The verification of the sensome genes was defined as genes with translated proteins that are located in the cell membrane (not secreted) and function to sense extracellular signals. In summary, this approach resulted in the selection of genes coding for proteins that are expressed at the cell membrane with a receptor-like function that are highly and specifically expressed by microglia. To determine orthologs, we manually curated all genes using the mouse genome information (MGI), Ensemble, and National Center for Biotechnology Information (NCBI) databases, and analyzed the similarities in function between orthologs using the GO function evidence codes [38,39].

### 4.4. Data Visualization

Heatmaps were generated using the gplots (version 3.01) heatmap.2 function in R based on the rlog values for each gene. Venn diagrams were generated using the draw.pairwise.venn function in the VennDiagram R package (version 1.6.18). The mouse to human expression data were extracted from the publication of Gosselin et al. and plotted using the ggplot2 (version 2.2.1) package [16]. The circular layout plots were generated using the circlize R package (0.4.9) [40].

## Figures and Tables

**Figure 1 ijms-22-01495-f001:**
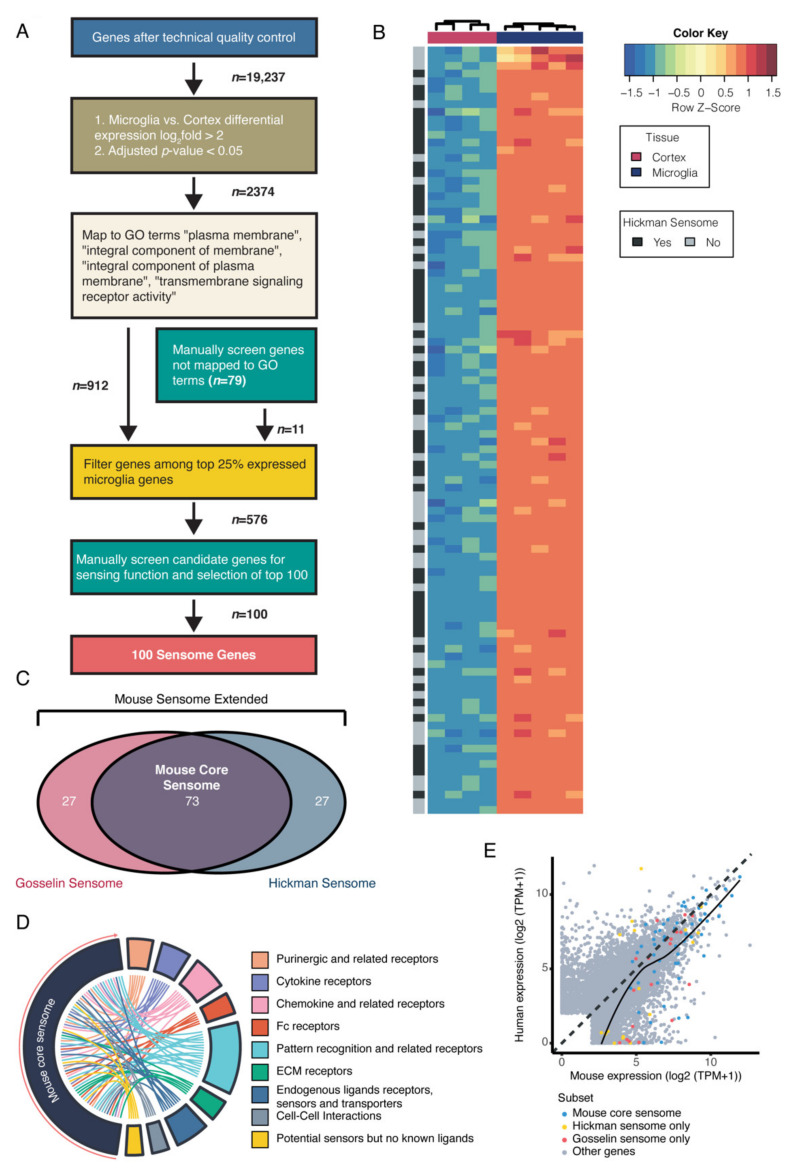
Identification of microglial mouse sensome shows major overlap between two independent datasets. (**A**) Flow chart showing selection of genes from the Gosselin et al. mouse datasets to identify the *n* = 576 candidate sensome genes. First, technical quality control was performed and genes with a low number of reads per sample were excluded. Second, differential expression analysis with a cut-off of log_2_fold >2 and *p*-adjusted value <0.05 was used to select genes significantly expressed by microglia. Third, using gene ontology (GO) annotation, we selected genes with association to “plasma membrane”, “integral component of membrane”, “integral component of plasma membrane”, and “transmembrane signaling receptor activity” (genes not mapped were manually curated). Fourth, the top 25% expressed genes were selected as candidate sensome genes. Finally, from the candidate sensome genes, the top 100 were manually verified using the criteria that these should express genes that translate in proteins that are present in the cell membrane and have a receptor-like function. In summary, these genes are highly expressed and specific to microglia and their respective proteins are located in the plasma membrane, where it has a receptor-like function. (**B**) Heatmap showing the first 100 sensome candidate genes (left to right) ordered by differential expression between cortex and microglia (low to high), with the upper row showing if the gene is present in “Hickman et al.”. (Appendix A). (**C**) Venn diagram showing the overlap between the top 100 Gosselin and Hickman mouse sensome genes (Appendix A). This total set is termed the “mouse sensome extended” and the shared set is called the “mouse core sensome”. (**D**) Circ plot showing the components of the 73 mouse overlapping sensome genes including the distribution of genes over the assigned subgroups. The genes are ordered based on log_2_fold differential expression (DE) extracted from the Gosselin dataset (microglia to cortex). (**E**) Scatterplot displaying the genes of the extended and overlapping mouse sensome, illustrating a minor enrichment of genes that are higher expressed in mice compared with humans. ECM, extracellular matrix.

**Figure 2 ijms-22-01495-f002:**
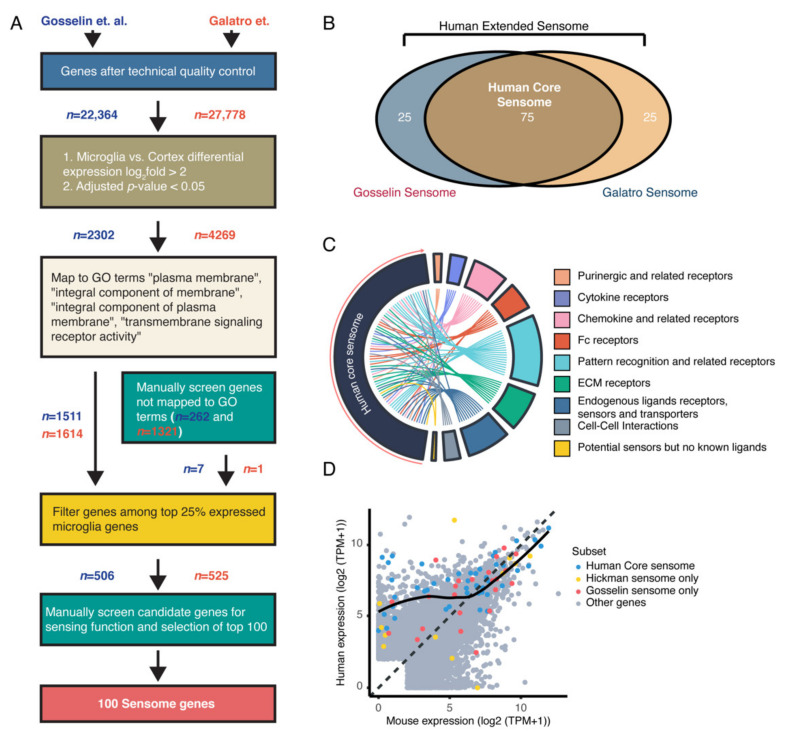
Identification of microglial human sensome. (**A**) Flow chart showing the selection of candidate genes using the Gosselin et al. human and Galatro et al. human microglial RNAseq datasets. A similar approach as in the mouse sensome selection was applied to finally select genes with translated proteins located in the cell membrane and that have a receptor function that are highly and specifically expressed by microglia. (**B**) Overlap between the Gosselin and Galatro mouse sensomes displayed in a Venn diagram, showing majority of genes from dataset overlap as candidate sensome genes. The total set is termed the “human sensome extended” and the shared set is called the “human core sensome” (Appendix A). (**C**) Human overlapping genes and assigned subgroups shown using Circ plot. (**D**) Distribution of human sensome genes showing a more evenly distributed over preferentially human and mouse expressed genes.

**Figure 3 ijms-22-01495-f003:**
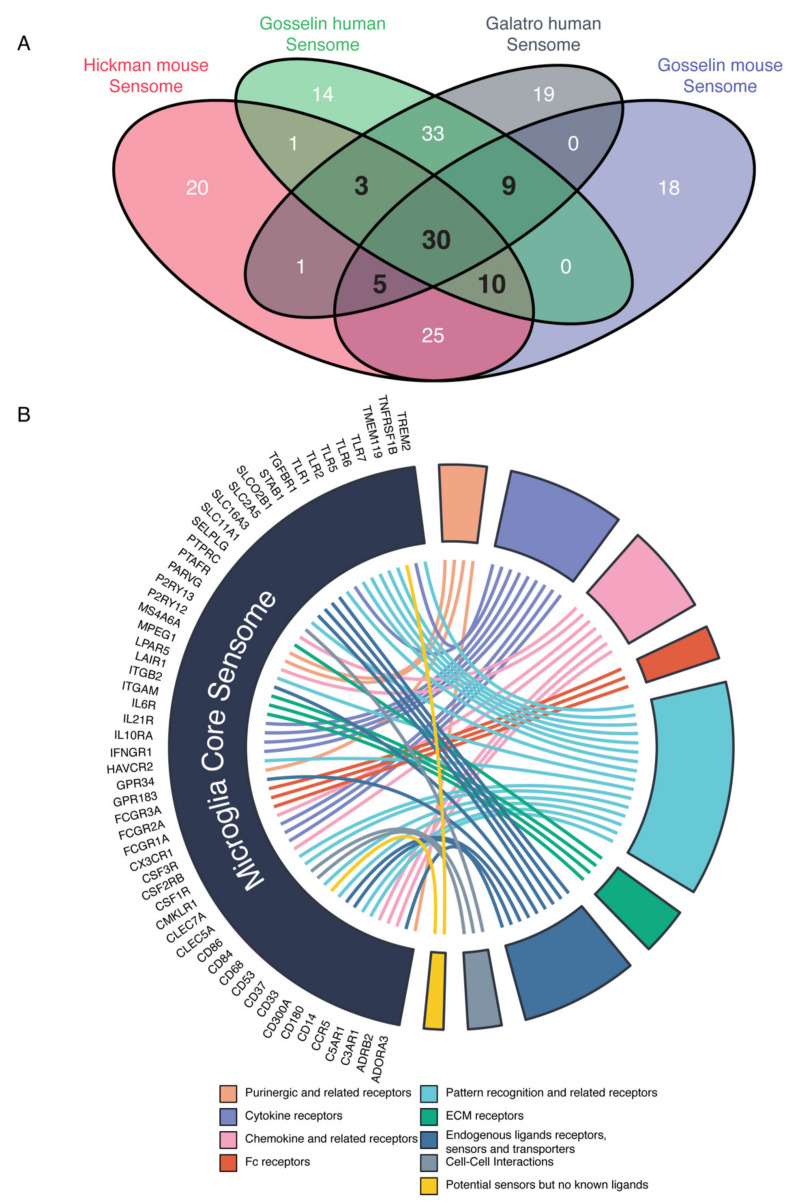
Overlap between mouse and human sensome reveals microglia core sensome. (**A**) Overlap between human and mouse sensome genes. We defined the microglia core sensome as genes present in three out of four sensome datasets (in bold). (**B**) Circ plot illustrating the functional subgroups of the microglia core sensome genes as identified by the shared expressed sensing genes in human and mouse microglia.

**Figure 4 ijms-22-01495-f004:**
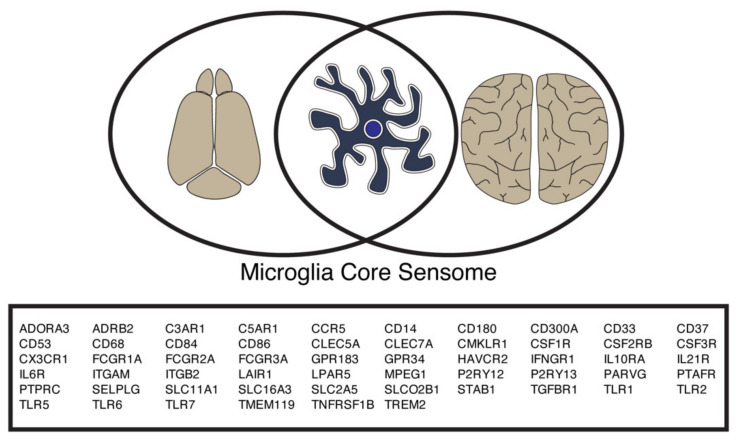
The mouse sensome and human sensome share similarities and differences. Overview of the microglia core sensome genes as identified by the overlap of human and mouse microglial sensomes.

**Figure 5 ijms-22-01495-f005:**
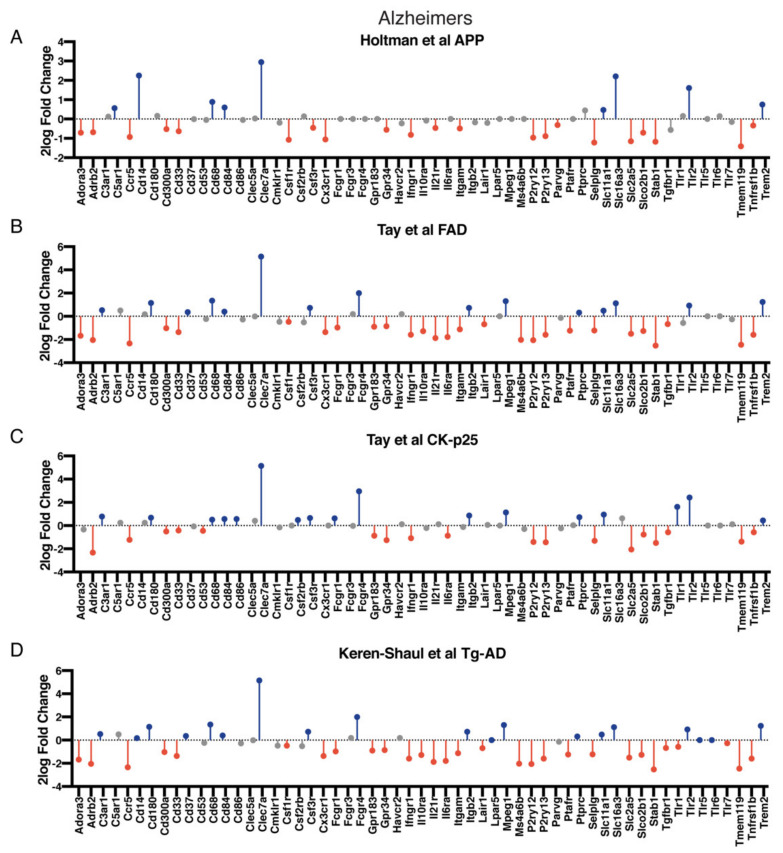
Microglia core sensome expression in Alzheimer’s disease (AD). (**A**) Two-log fold change of microglia core sensome genes in microglia in AD mouse model (APP) [12]. (**B**) Microglia core sensome gene expression in 5xFAD mouse model [13]. (**C**) CK-p25 mouse model gene expression of microglia showing changes in microglia core sensome [13]. (**D**) Differential gene expression analysis of single cell microglia data comparing homeostatic microglia versus disease-associated microglia (DAM) in an AD mouse model (Tg-AD) showing changes in microglia core sensome [4]. Red bars display gene significantly upregulated, blue bars represent gene significantly downregulated (detailed expression data in Appendix A).

## Data Availability

Data available from original sources as referenced.

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
