# Peer review of "Comparative Analysis Identifies Similarities between the Human and Murine Microglial Sensomes"

_ijms, 2021, doi:10.3390/ijms22031495_

Round 1

Reviewer 1 Report

The manuscript under review entitled “Comparative analysis identifies similarities between the human and murine “microglial sensomes” reports on an investigation carried out through a database analysis which refers to the interesting concept of a “core microglia sensome” shared between mouse and man. The Authors aim was, in fact,  to validate and extend the yet published mouse microglia sensome contextually defining the human microglia sensome, and determing which sensing genes could be highly expressed in both species. The Authors successfully identified a set of 57 genes that seems important for specific microglial sensing in both human and mouse. They termed this set of genes the “microglia core sensome”. This is a good work since opens a new investigative scenario useful for future studies. Understanding the changes that the microglia sensome can undergo in the course of various inflammatory pathologies can be of great use in order to identify new therapeutic targets for the development of new drugs. This work will certainly be of great interest to the readers of this journal and, in my opinion, it can be considered for a possible publication after some minor revisions.
Best regards.

Comments for the Authors
- There are some typographical errors in the manuscript. Please the Authors should check correct typos.
- The authors propose that the monitoring of some microglia sensome genes may be useful to understand the evolution of some neurodegenerative diseases such as Alzheimer's and ALS. However, they make no reference to Parkinson's disease. Could they fill this gap? Is there any information about it? If so they should add them in the text.
- Supplementary Materials Section: The table S5 looks broken. In some places it is not completely readable. Please check and modify if necessary. In this section, moreover, reference is made to Fig S1, Fig S2, Fig S3, Fig S4, Fig S5 and Fig S6. The figures, however, do not seem to be present in this section. Please check and add if necessary.

Author Response

Comments for the Authors

- There are some typographical errors in the manuscript. Please the Authors should check correct typos.

We thank the reviewer for their thorough review of our manuscript and appreciate all their constructive comments. We proofread the manuscript to check for typos. We have now corrected the typos in the new manuscript.

- The authors propose that the monitoring of some microglia sensome genes may be useful to understand the evolution of some neurodegenerative diseases such as Alzheimer's and ALS. However, they make no reference to Parkinson's disease. Could they fill this gap? Is there any information about it? If so they should add them in the text.

Unfortunately, we were unable to find sufficient information into the role of microglia in the progression of Parkinson’s disease. We have included this in the discussion:

In this analysis into the different diseases, Parkinson’s disease was not included. This disease model has not been studied to such an extent compared to the other neurodegenerative disease analyzed here. Future studies into Parkinson’s disease and specifically into microglia can shed light into the role of these cells and how they are affected during disease progression.” (Page 11, Line 366-370)

- Supplementary Materials Section: The table S5 looks broken. In some places it is not completely readable. Please check and modify if necessary. In this section, moreover, reference is made to Fig S1, Fig S2, Fig S3, Fig S4, Fig S5 and Fig S6. The figures, however, do not seem to be present in this section. Please check and add if necessary.

During the submission process these figures were not included in the reviewer’s version of the manuscript, we have now included these figures in the final manuscript. These figures can be found after the references.

Reviewer 2 Report

This manuscript seeks too address the differences between human and murine microglial sensomes. It provides some timely analysis of existing datasets to provide conclusions regarding similarities across species. The data indicates overlapping core genes that regulate microglial function across species. It is well written and the analysis is presented in an accessible manner. I recommend that the article be published. 

Author Response

This manuscript seeks too address the differences between human and murine microglial sensomes. It provides some timely analysis of existing datasets to provide conclusions regarding similarities across species. The data indicates overlapping core genes that regulate microglial function across species. It is well written and the analysis is presented in an accessible manner. I recommend that the article be published.

We thank the reviewer for the time invested in the reviewing of our manuscript. We appreciate the feedback and thank the reviewer for recommending our manuscript to be published.

Reviewer 3 Report

Investigations on the microglial sensome are of interest and the dissemination of data on this topic should be favored. However, further studies will have to confirm that the observed changes in the expression of the sensome genes correspond to parallel changes in the synthesis of the protein molecules that they encode. As well as future studies in this field will have to focus on the regional differences that exist in the various  microglial populations

Author Response

Investigations on the microglial sensome are of interest and the dissemination of data on this topic should be favored. However, further studies will have to confirm that the observed changes in the expression of the sensome genes correspond to parallel changes in the synthesis of the protein molecules that they encode. As well as future studies in this field will have to focus on the regional differences that exist in the various microglial populations.

We thank the reviewer for their thorough review of our manuscript. We have addressed the editing of English language and style in the new manuscript. As discussed, we acknowledge that future studies are necessary to confirm if the observed changes in the microglia sensome gene expression correspond to protein expression. We now also include the caveat that future studies should also take into account the spatial differences and the different microglia subpopulations as are now discovered using single cell technologies.